# The Roles of Depression, Life Control and Affective Distress on Treatment Attendance and Perceived Disability in Chronic Back Pain Sufferers throughout the Duration of the Condition

**DOI:** 10.3390/ijerph20196844

**Published:** 2023-09-27

**Authors:** Humberto M. Oraison, Daniel Loton, Gerard A. Kennedy

**Affiliations:** 1Institute for Health & Sport, Victoria University, Melbourne, VIC 3000, Australia; dloton@unimelb.edu.au; 2Centre for Wellbeing Science, The University of Melbourne, Melbourne, VIC 3010, Australia; 3School of Science, Psychology and Sport, Federation University, Ballarat, VIC 3842, Australia; g.kennedy@federation.edu.au; 4Institute for Breathing and Sleep, Austin Health, Heidelberg, VIC 3084, Australia; 5School of Health and Biomedical Science, RMIT University, Bundoora, VIC 3083, Australia

**Keywords:** depression, life control, affective distress, chronic back pain, disability, treatment

## Abstract

The aims of this study were to examine psychological factors that predict treatment seeking and disability over the total duration of experiencing back pain. A sample of 201 adults experiencing chronic back pain was recruited through health professionals and completed the Depression, Anxiety and Stress Scale (DASS), the Oswestry Back Pain Disability Questionnaire (ODQ), the McGill Pain Questionnaire (MPQ) and the life control and affective distress variables of the West Haven–Yale Multidimensional Pain Inventory (WHYMP), and participants disclosed the number of treatment sessions attended over the course of the illness. Depression, life control and affective distress were tested as indirect predictors of disability severity that were mediated by treatment attendance. Each unit increase in life control predicted attending nearly 30 more treatment sessions, each unit increase in affective distress predicted attending 16 fewer treatments and each unit increase in depression predicted 4 fewer treatments, together explaining 44% of variance in treatment seeking. The effects of life control and affective distress on disability were explained by treatment attendance, whereas depression retained a direct effect on disability. Treatment attendance had an effect on disability. The findings show that participants with lower life control and higher affective distress and depression had higher levels of pain and disability, in part due to due to their treatment-seeking behaviour.

## 1. Introduction

There is a well-established positive relationship between chronic back pain and psychological distress [1,2,3,4]. Underpinning this relationship is a complex series of interacting factors, which have plausible theoretical explanations of how these variables may have both direct and indirect effects on patient outcomes [5]. Paradoxically, patients who suffer more severe pain tend to attend fewer treatments; Ref. [2] pooled results from seven studies that suggested that despite increases in the prevalence of lower back pain (LBP) and in treatment options, less than 60% of sufferers sought treatment for this complaint. Research has therefore moved from investigating the direct effect of chronic back pain on perceived disability to exploring mechanisms that may explain variation in treatment seeking, adherence and patient outcomes.

Among these possible mechanisms, psychological variables are paramount, as they have theoretically plausible effects on the likelihood to both seek and adhere to treatment, as well as directly exacerbating perceived disability. Overall, there is very limited research establishing the role of psychological factors in the direct and indirect outcomes of chronic back pain (CBP), via treatment seeking, across the entire course of a CBP diagnosis [6].

### 1.1. Depression, Treatment Seeking and Perceived Disability

Chronically ill patients suffering from depression report a higher number of medical symptoms [4,7]. Goldstein [8] found that depression was associated with poorer adherence to self-care regimes (e.g., diet, exercise, medication compliance, etc.) and with complications in chronic medical illnesses and increased symptoms [8]. There is widespread agreement about the comorbidity of depression and chronic pain [1,4,7]. Pincus and Morley [9] proposed a model to describe the relationship between depression and pain. In this model, instead of assuming a duality of the perceptions of pain and depression, there is one problem (pain) that is perceived both physically and affectively. Pincus and Morley suggested that chronic back pain sufferers had biased schemas that could be associated with the presence of psychological disorders. They concluded that psychological distress, pain severity and catastrophising interact to produce functional disability, suggesting that treatments should focus on the cognitive processes of pain biases. Keeley et al. [6] found that depression and beliefs related to pain and social stress were predictors of lower levels of attendance for treatments.

While depression correlates with pain, untreated depression can interfere with the effectiveness of treatments for chronic pain [10]. Depression has been found to adversely affect the efficacy of pain treatments [11] and cause “low adherence to treatments” [12]. Silva et al. [13] found depression to be a predictor of functioning and suggested that a measure of depression together with global pain intensity be included in routine clinical assessments.

### 1.2. Life Control, Treatment Seeking and Perceived Disability

Life control can be defined as the individual’s ability to understand internal and external environments, the ability to influence his or her life course, the meaningfulness of his or her lived experiences and life satisfaction levels [14]. Kerns et al. [15] linked perceptions of control to confidence and problem-solving abilities. Utilising the Problem-Solving Inventory (PSI, third subscale, [16]), they found significant correlations between problem solving and pain, disability and depression scores among chronic pain sufferers. In particular, personal control represented greater perceived problem-solving abilities, and this was negatively correlated with pain, disability and depression. Härkäpää et al. (1991) [17] found that healthy locus of control beliefs were associated with successful outcomes in treating back pain. Patients with stronger internal beliefs benefited from treatment and completed back exercises. In addition, they found that symptoms of psychological distress were significantly associated with poorer adherence to back exercises. Life control perceptions may affect individuals’ ability to judge processes and even time changes [18]. In a later study, Härkäpää et al. [17] found that health optimism and control beliefs were significant predictors of improvements in functional capacity. These studies used depression scales and the Health Optimism Scale, which include items about locus of control beliefs and pain beliefs [17]. Patients with CBP reported lower general satisfaction with treatments and lower quality of life in a Korean study [19]. Life control mediated the relationship between pain severity and physical impairment. Perceived life control was found to be one mechanism by which PTSD symptoms and pain severity were indirectly associated, with psychosocial impairment indirectly related to physical impairment [20].

### 1.3. Affective Distress, Treatment Seeking and Perceived Disability

Affective distress is a biologically based process that includes a cognitive appraisal process and has a state of action readiness to respond. Affective distress has been called a central and organising symptom of depression [21]. Affective distress includes anxiety and depression, and both have similar responses as pain in the physiological system (with shared neurological pathways). Affective distress may be a central symptom of depression, appearing as the environment for negative cognitive symptoms that lead to dysfunctional beliefs and pessimistic information processing [21]. Pain perception may increase with higher levels of affective distress and anxiety. Affective distress in Turkish cancer patients impacted their pain scores more significantly than pain intensity [22]. In addition, these authors found that managing emotional/affective distress may prove valuable for controlling pain and improving satisfaction with treatment for cancer patients. A successful treatment for cancer pain would include the patient’s own perception and conceptualisation of their pain experience [22]. Perceived life chaos is highly correlated with anxiety and depression and has been found to be associated with medication adherence [23,24]. Greater affective distress was found to moderate the effect of the belief in treatment effectiveness on adherence for chronic conditions. Affective distress and life control scores significantly predicted pain intensity and pain unpleasantness levels among patients with orofacial pain [25].

Although no relationship between pain intensity and disability was established, a clear association was found in relation to negative cognitions reducing effort levels during physical activity and increasing avoidance due to feelings of fear. It has been argued that the cause of disability in all pathologies is not the injuries or disease but the actual pain they produce [26].

### 1.4. Aims and Hypotheses

There is limited research about factors influencing attendance for treatment and whether there are positive outcomes of treatment [27]. We have established that there are plausible theoretical explanations as to how the three psychological variables investigated in this study can exert both direct and indirect effects on CBP patient outcomes via treatment attendance. Therefore, the aims of this study were to test the direct and indirect effects of these psychological factors on perceived disability over the entire course of a CBP diagnosis experience (the experience of back pain beyond a six-month period). In line with the suggestions of Pincus and Morley [9], we hypothesise the following:

**H1:** *Depression (positive), life control (negative) and affective distress (positive) will have a significant direct effect on perceived disability, controlling for age, gender, long-term pain duration and severity*.

**H2:** *Depression (negative), life control (positive) and affective distress (negative) will have a significant indirect effect on perceived disability via treatment attendance, controlling for age, gender, long-term pain duration and severity*.

## 2. Method

### 2.1. Participants

Two hundred and one participants were recruited with the assistance of health professionals, who distributed questionnaires amongst their patients. The inclusion criteria were that participants must have experienced chronic or recurrent back pain for more than three months [28] and be 18 years or older. There were 89 men and 112 women, all of whom lived in metropolitan Melbourne, Australia. The participants were aged between 19 and 88 years and the mean age was 47.18 years (*SD* = 13.44). Most participants reported having experienced pain for longer than 24 months (72.6%). One fifth (20.9%) of participants reported experiencing pain for between 6 and 24 months.

### 2.2. Materials

#### 2.2.1. Depression Anxiety Stress Scales (DASS)

The DASS consists of three self-report scales that have been designed to measure the negative emotional states of depression, anxiety and stress. Each of the 3 DASS-42 scales includes 14 items representing the dimensions of depression, anxiety and stress [29]. Each of the 42 questions is scored on a 4-point scale ranging from 0 (“Did not apply to me at all”) to 3 (“Applied to me very much, or most of the time”). Scores for depression, anxiety and stress are calculated by summing the scores for the relevant items. The internal consistency for the DASS 21 was high, with Cronbach’s alphas at 0.94 for depression, 0.87 for anxiety and 0.91 for stress.

#### 2.2.2. McGill Pain Questionnaire (SF-MPQ)

The pain rating index of the McGill Pain Questionnaire (MPQ) [30] was utilised to measure the severity of pain (ranging from 0 = no pain to 5 = excruciating). This subscale demonstrated an intraclass correlation coefficient of 0.75, and its internal consistency reliability was also high, with a Cronbach’s alpha of r > 0.75.

#### 2.2.3. Oswestry Low Back Pain Questionnaire

The Oswestry Low Back Pain Disability Questionnaire (ODQ) can be used to assess patients with low back pain by determining its impact on activities of daily living (i.e., sitting, standing and walking) [31]. This questionnaire has 10 items with a range from 0 to 5 for each question, and scores are added and range from no disability (0–4) to completely disabled (35–50). The ODQ is a widely used instrument which has a strong internal consistency (alpha = 0.85).

#### 2.2.4. Life Control and Affective Distress (West Haven–Yale Multidimensional Pain Inventory, WHYMPI)

This study utilised two subscales from the West Haven–Yale Multidimensional Pain Inventory (WHYMPI/MPI), rating (0–6) patients’ perceived life control (two items), and affective distress (three items). Scores for each scale were added and divided by the number of items. Kerns et al. [32] demonstrated that the internal reliability coefficients of all WHYMPI scales range from 0.70 to 0.90.

#### 2.2.5. Demographic Data Collection Questionnaire (DDCQ)

A 12-item questionnaire was designed for the collection of demographic data and included variables considered to have an impact on the experience of chronic low back pain, including type of health professional and number of sessions attended to treat back pain, as well as gender, age, intensity and duration of pain experience.

The treatment attendance variables ranged from 0 to 450 visits. We examined the types of professionals, and many were homogenous allied health professionals (physiotherapists, acupuncturists, osteopaths, remedial masseurs, hydrotherapists, etc.).

### 2.3. Procedure

A total of 450 self-addressed, reply-paid (return postage provided) envelopes containing the study details, a consent form and the battery of questionnaires were distributed through allied health professionals. A letter of invitation was provided to health professionals to give to their patients. (Five of each of the following professional specialties were contacted: chiropractors, physiotherapists, osteopaths and acupuncturists.) The letter included information about the study as well as the requirements of being involved in the recruitment process and how to deliver the questionnaires to patients. Health professionals consented to be part of the recruitment process by signing and returning a consent form. The data collected in the questionnaires were analysed utilizing the Statistical Program for Social Sciences Version 22 (SPSS). The response rate was 201/450 = 44.66%. Each participant signed a consent form after reading a letter of invitation to participate with information about the research as well as details to access psychological support in the case of any psychological disturbance as a result of their involvement in this study.

As the unique focus of this study was on the effects of psychological variables that may influence treatment attendance and, indirectly, disability outcomes over the total illness duration, the pain duration variable was examined closely. As can be seen in Table 1, the vast majority of participants exhibited back pain for longer than 6–24 months, which meets the cut-off criteria for chronic back pain as per Chou [33]. The vast majority of participants had experienced chronic back pain for over 2 years, followed closely by 6 months–2 years, with only 10 participants experiencing pain for less than 6 months. All participants were retained for the analysis.

### 2.4. Variables

Demographic variables were collected from the purposely devised demographic questionnaire (gender and age). Pain variables were collected from the McGill Pain Questionnaire (pain rating index severity: none, mild, moderate and severe) and demographic questionnaire (duration of pain: 0–3 months; 3–6 months; 6–24 months and over 24 months). Psychological variables were extracted from the DASS 21 (depression, moderate and severe) and the WHYMPI (perceived life control and affective distress). As the study focussed on the potential mechanisms of depression, life control and affective distress on CBP via treatment attendance and prior studies have found differences in CBP severity and psychological variables across gender and age and pain severity, we included all as control variables in the substantive analysis.

### 2.5. Statistical Analysis

Following data cleaning, checks for the initial conditions of mediation were undertaken using Pearson’s correlation coefficients. The substantive analysis then consisted of a path model estimating the indirect effects of depression, life control and affective distress on disability, via treatment attendance, while controlling for age, gender, total pain duration and pain severity (Figure 1).

Model fit statistics were examined against commonly applied benchmarks of the Tucker–Lewis Index (TLI), and the comparative fit index (CFI) of >0.95, root-mean-square error of approximation (RMSEA) with 90% CI and standardised root-mean-square residual (SRMR) of >0.05, as well as the χ^2^ comparison to a baseline model, indicated a close model–data fit [34,35]. Significances of indirect effects were evaluated using bias-corrected and accelerated confidence intervals with 10,000 bootstrapped samples at an alpha level of 0.95 (the effect was deemed significant if these confidence intervals did not include zero). Unstandardised effects were preferred for their ease of interpretation, especially in relation the number of lifetime back pain treatments attended, which had a larger range than the other variables in the model. Correlation coefficients were generated using IBM SPSS Version 26, and the path model was estimated using the R package lavaan (version 0.6–10); [36].

## 3. Results

Descriptive statistics for all measures utilised are presented below. Means and standard deviations of the measures are presented in Table 2, and Pearson’s correlation coefficients are presented in Table 3.

Although guidelines vary as to the required sample size in path or structural equation models [37] our sample size was adequate based on the general rule of five or more participants per model parameter that was estimated in a CFA framework [37]. Missingness was negligible at <1% of the sample and was handled with list-wise deletion. Mediation conditions were present, with depression, life control and affective distress all demonstrating correlations with the number of treatment sessions attended and perceived disability.

The substantive analysis consisted of the path model estimating the indirect effects of depression, affective distress and life control on attendance for treatments and on perceived disability. Some model fit statistics were adequate, with χ^2^ = 11.374(4), *p* < 0.0.023, SRMRs of 0.03, and CFIs of 0.95, while others were poor, with higher RMSEA point estimates and wide 90% confidence intervals (RMSEA = 0.101 [0.034, 0.173]). A significant chi-square was expected due to the sensitivity of this statistic to sample size but also partly due to the much larger range of the variable of treatments attended, compared with the ranges of the other variables. These are the likely causes of the higher-significance chi-squares and higher RMSEAs. Given these results and that the substantive model was primarily generated based on theories explaining the connection between these variables, we proceeded to interpret the path model results without testing alternative model specifications.

### Direct and Indirect Predictors of Treatment Attendance and Disability

Together, depression, affective distress and life control explained a substantial amount of variance in the number of treatments attended (*R*^2^ = 0.44), which, together with all predictors, explained a substantial amount of perceived disability (*R*^2^ = 0.40).

The unstandardised results are presented in Figure 2. Each increase of a point on the depression scale predicted attending nearly 4 fewer treatments (b = −3.87 (0.90) [−5.78, −2.20]), and for affective distress, there was a much larger effect of nearly 17 fewer treatments attended (b = −16.68 (7.01) [−30.35, −2.68]). Life control, however, had the largest effect on treatment attendance, with each increase predicting nearly 30 more treatments attended (b = 29.38 (7.53) [15.52, 44.92]).

In turn, the number of treatments attended had a small inverse effect on perceived disability (b = −0.01 (0.00) [−0.02, −0.01]). In terms of evaluating the indirect effects, depression demonstrated partial mediation, as the direct connection to disability remained significant (b = 0.32 (0.07) [0.15, 0.51]), whereas life control (b = 0.55 (0.56) [−0.61, 1.61]) and affective distress (b = 0.30 (0.56) [−0.83, 1.42]) had no significant effect on disability after taking into account their indirect effects via treatment attendance.

Bias-corrected and accelerated confidence intervals for the indirect effects supported a small significance of the indirect effects, but it often bordered on zero at the lower bound. Depression had a small indirect effect on disability (b = 0.04 (0.02) [0.01, 0.10]), with a larger inverse effect on life control (b = −0.38 (0.18) [−0.85, −0.09]) and a similar indirect effect on affective distress (b = 0.21 (0.08) [0.04, 0.56]). While the effect of treatment attendance on disability appears small, as the treatments attended were across the entire duration of experiencing back pain, they had a large range. Hence, for each treatment session attended, to see a small significant decrease in disability can quickly become meaningful when the large differences across individual treatments attended are taken into account. Nonetheless, the total indirect effect was small relative to the total effect, making up 2.67% of the total effect.

## 4. Discussion

This study directly tested three promising candidates that may explain the mechanism behind treatment attendance by testing their direct and indirect effects in a sample of CBP sufferers over the entire duration of their condition. All three psychological variables (depression, life control and affective distress) predicted disability, as did treatment attendance. Each treatment attended predicted lower disability, indicating the importance of treatments and rehabilitation in the experience of CBP. In addition, the fact that this was held over the course of the entire illness duration is important and unique to this study (controlling for illness duration in the model).

### 4.1. Depression

The findings were consistent with previous research identifying depression as having a direct and indirect effect (via treatment attendance) on disability. Depression may act as a demotivator for attending treatments [38], as it is related to fearful cognitions [6] and associated with reduced physical activity and increased fear avoidance [27] and catastrophizing [39], all contributing to lower levels of attendance for treatments. In addition, the theory of biased schemas for back pain sufferers [9] may contribute to explaining the role of depression as a treatment-seeking deterrent. Given the findings of our study and the evidence in the existing literature, it is clear that psychological distress plays a major role as a deterrent in treatment seeking for CBP. Depression causes cognitive distortions, creates avoidance and fear and reduces motivation levels as well as causing psychosomatic symptoms that further aggravate the experience of chronic pain [39].

Depression also retained a direct effect with disability, irrespective of treatment attendance. This finding is in agreement with the literature [1,7] with some authors attributing this relationship to the interaction between pain and depression, while others have advocated for a new schema that considers pain and depression as “two perceptions of the same problem” [9]. The same schema arising from depression that may make a patient less likely to seek treatment, as they evaluate a lower likelihood of treatment success, may also act directly on evaluations of disability, with the negative schema exacerbating disability evaluations. Despite a great deal of agreement regarding the relationship between depression and chronic pain, there is little agreement in the literature in relation to causality. If depression and physical disability are indeed two sides of the same coin, one physical and one emotional, it may be adequate to consider them as one variable of a new concept of global disability. Most of the literature and most existing questionnaires and evaluation tools utilised to measure disability mainly focus on the physical impacts, as well as impairment and loss of functionality, but do not measure psychological factors.

Díaz-Aristizabal et al. [40] found that more than 50% (*r* = 0.54) of individuals who reported disability had clinical depression as well. Melzack [31] suggested that chronic pain sufferers develop frustration due to the duration of their symptoms and feel misunderstood by the medical profession because their pain continues in the absence of a significant physical injury. Others go further, describing a mutually enforcing relationship between depression and chronic pain disability [7,30].

These findings and previous research have led to support of the idiosyncrasy of CBP as a medical condition, with psychological distress as an integral part of the disability that is interwoven into physical and emotional systems, leading to loss of functionality and impairment in both areas. The presence of psychological distress in the experience of CBP acts both as a barrier in motivational terms, negatively impacting recovery, and as fuel for catastrophizing a pessimistic outcome [39] Further investigation and exploration on how catastrophizing may contribute to the chronicity and entrenchment of back pain beyond the reasonably expected period of recovery of an initial physical injury is required.

### 4.2. Life Control

Although the main source of psychological information was the Depression, Anxiety and Stress Scale (DASS, [29]) life control and affective distress (WHYMPI, [30] were chosen as additional independent variables in the analyses. Given that life control measures coping strategies, it has proven to be an invaluable source of information, considering that coping has been identified as an essential element of psychological resilience for chronic pain sufferers. Life control had an indirect effect on disability, which may indicate that individuals with high problem-solving abilities and who feel in control of their lives’ events may also be more able to engage in treatments [15,17]. These findings coincide with Härkäpää et al. [17] in reaffirming that control impacts functional capacity and with Choi et al. [19] as perceived life control was also found to be a mechanism to account for psychosocial impairment [20].

### 4.3. Affective Distress

Affective distress also had an indirect effect on disability. This finding is in line with previous research [22,23,24,25]. Perceived life chaos can be equated to affective distress [24] and may explain the impacts of this effect on treatment adherence [23] due to a diminished confidence in the effectiveness of the treatments. Affective distress may impact the sufferers’ perceptions of their pain experience and their ability to implement changes through treatments.

While depression retained a direct effect on disability, the effects of life control and affective distress were completely explained by treatment attendance. One of the possible explanations for these differences may lay in the fact that the measure of depression may have a wider spectrum of symptomatology and impacts. Life control scores may reflect feelings like hopelessness and even a distorted perception in relation to the lapse of time (slower) [18]. In the same manner, affective distress may be a more selective and limited indicator, focussing on negative self-knowledge and negative evaluation processes which may have a marked impact on the CBP sufferer’s beliefs in the efficacy of treatments and therefore make him or her more unwilling to engage [21].

Receiving treatment is in fact the key in preventing the transition from acute to chronic pain, as mentioned earlier [41]. While the experience of acute pain is mainly determined by an identifiable source of pain, chronic pain has a social and psychological idiosyncrasy that overtakes the role of physical pain. This study focussed on highlighting the role of psychological distress as a core element of the experience of CBP and disability, irrespective of causality and direction. Individuals suffering from CBP are not only limited by physical dysfunction but also, and perhaps more importantly, limited by psychological distress.

There are grounds to advocate for a new concept of disability measuring a global loss of functioning given the evidence of how emotional and physical symptoms interact to impact the back pain sufferer. Having a clearer understanding of the mechanisms leading to treatment engagement would give clinicians better opportunities to treat negative cognitive schemas that lead to lower engagement and higher disability. Psychological indicators have been established as significant and important determinants of the level of attendance for treatments.

These treatments (or lack thereof) were also associated with higher levels of disability. The contribution of this research lays in exposing the role of treatments in relation to psychological factors and disability. The consistent nature of these findings demonstrated that participants who were psychologically distressed not only had higher levels of disability but also attended fewer treatments and had longer durations of back pain. Further research is required to focus on the relationship between treatments and the transition from acute to chronic back pain.

## 5. Limitations

This study had a participation rate of less than 50%. This low participation may be attributed to recruitment issues, as the questionnaires were given to health professionals to pass on to their patients. Furthermore, the range in the number of sessions attended appears to be very high, which, given that the study occurred over the course of the illness and some people would have experienced CBP for longer than others, is somewhat expected. This large range highlights how even a small effect of treatment attendance on disability can quickly compound with additional treatments over the course of the illness, resulting in substantially less disability. Control variables were used to remove the effect of illness duration on the outcomes; however, future studies could target a more precise sample of people who have experienced longer term back pain. This study did not collect data on participants’ psychiatric diagnoses or pharmacological treatments; further studies may consider these two aspects and include them for further investigation. In addition, the study relied completely on self-reported data, which must be considered while evaluating the findings.

## 6. Conclusions

In conclusion, chronic back pain appears to have a biopsychosocial presentation, with a clear indication that psychological factors have an impact on treatment engagement and disability. The level of participation in treatments was influenced by each person’s own psychosocial reality rather than the level of pain experienced by participants [42], extending the transition from a short episode of acute pain related to a physical injury to a long-term and chronic experience of pain even when the physical impact of the injury had resolved.

For individuals already in a chronic pain stage, there should be an emphasis on treating their psychological distress, given the fact that psychological distress can not only magnify their actual pain but can also interfere with their cognitive ability, preventing them from engaging in physical rehabilitation. The literature researched and the findings of this study clearly identify the strong presence of psychological distress as a pivotal component of chronic pain. This knowledge may lead clinicians to include psychological treatments with the physical interventions for chronic back pain in order to ensure optimal adherence to treatments. Despite an exhaustive search of the current literature, there is no current tool or questionnaire specifically designed to encompass all areas involved in chronic pain.

## Figures and Tables

**Figure 1 ijerph-20-06844-f001:**
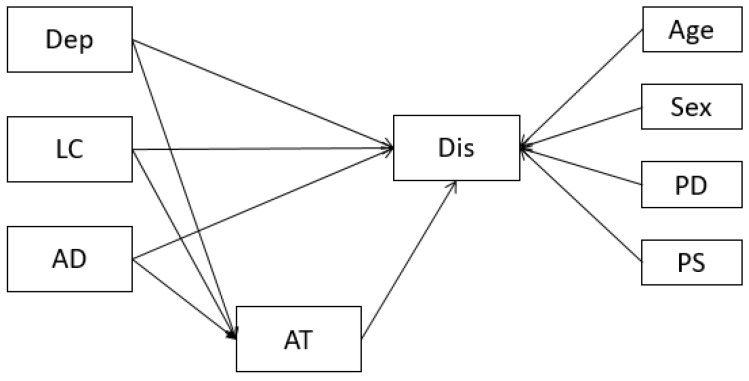
Hypothesised model with depression, life control and affective distress indirectly predicting disability via treatment attendance. Note: Dep = depression; LC = life control; AD = affective distress; AT = attending treatments; Dis = disability; PD = pain duration, total; PS = pain severity. Control variables are presented on the right-hand side of the model. Covariances between predictors are not displayed for brevity.

**Figure 2 ijerph-20-06844-f002:**
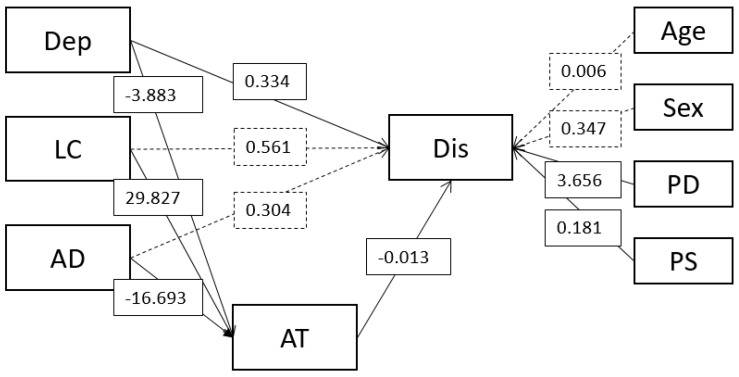
The full mediation model. Values depict unstandardised *β* weights. Note: Dep = depression; LC = life control; AD = affective distress; AT = attending treatments; Dis = disability; PD = pain duration, total; PS = pain severity. Control variables are presented on the right-hand side of the model. Dashed lines represent non-significant paths (*p* > 0.05). Covariances between predictors are not displayed for brevity.

**Table 1 ijerph-20-06844-t001:** Overall pain duration frequency (%) of the sample.

Duration	Count (%)
0–3 months	2 (1.11%)
3–6 months	8 (4.44%)
6–24 months	41 (22.78%)
24+ months	129 (71.67%)

**Table 2 ijerph-20-06844-t002:** Means and standard deviations for men and women by psychological indicators, number of treatment sessions received and disability scores.

Variable	Men	Women	t (df)	*p*
Life Control	4.58 (4.29)	4.74 (4.40)	−2.71 (178)	0.01
Affective Distress	3.35 (1.57)	2.87 (1.31)	2.21 (178)	0.03
Depression	16.00 (9.88)	14.34 (11.72)	1.02 (178)	0.01
No. of Sessions	118.77 (193.57)	191.36 (284.54)	−22.02 (156)	0.05
Disability (ODQ)	18.34 (8.80)	16.02 (10.57)	1.59 (173)	0.11

**Table 3 ijerph-20-06844-t003:** Pearson’s correlations between variables.

	No. of Sessions	Disability	Life Control	Aff. Distress	Pain Score	Depression
No. of Sessions	-					
Disability	−0.38 **	-				
Life Control	0.14	−0.25 **	-			
Affective Distress	−0.21 **	0.34 **	−0.70 **	-		
Pain Score	−0.11	0.32 **	−0.31 **	0.38 **	-	
Depression	−0.51 **	0.55 **	−0.51 **	0.58 **	0.35 *	-

** Correlation is significant at the 0.01 level (2-tailed). * Correlation is significant at the 0.05 level (2-tailed).

## Data Availability

The data that support the findings of this study are openly available in “Dryad” at https://doi.org/10.5061/dryad.cnp5hqc61.

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
