# Peer review of "The Roles of Depression, Life Control and Affective Distress on Treatment Attendance and Perceived Disability in Chronic Back Pain Sufferers throughout the Duration of the Condition"

_ijerph, 2023, doi:10.3390/ijerph20196844_

Round 1

Reviewer 1 Report

The topic of this study is very difficult, complex, and broad. It is not very easy to carry out research in this area. 

In my opinion, there are not methodologically well-set limits for the duration of the pain (according to which, for example, respondents who state that the duration of their pain is 6 months are included in the group?). The authors should explain this.

The presentation of results in the model is very clear. Multivariate statistics would be appropriate. The scientific soundness would increase. There is no conclusion in the article, just a mention in the Limitations section.

Author Response

Reviewer 1:

  1. In my opinion, there are not methodologically well-set limits for the duration of the pain (according to which, for example, respondents who state that the duration of their pain is 6 months are included in the group?). The authors should explain this.

We have now included a table breaking down the frequency counts and percentages for the back pain duration categories utilised in the study. These reveal the vast majority of participants had experienced chronic back pain, as per the definition of at least six months in AIHW, 2016a, or 3 months according to other criteria (Chou, 2010). Only 10 participants experienced back pain for less than 6 months. All participants were retained in the analysis. We have added a description of this variable, and explained how the vast majority of participants are suitable to the research question as they meet the criteria for chronic back pain.

  1. The presentation of results in the model is very clear. Multivariate statistics would be appropriate. The scientific soundness would increase. There is no conclusion in the article, just a mention in the Limitations section.

Thank you for the comment on the results section. To our understanding, we used multivariate statistics. We fitted a path model with multiple predictors, indirect effects and outcomes, including suitable control variables of variables identified in prior studies as important, on which the outcomes were regressed. A path model is multivariate statistics. The limitations discuss the lack of measurement error control in the form of a full structural equation model that models item-level measurement error, and we recommend future studies fit full structural equation models where sample size and data allows for this.

Similarly, we did not follow the meaning of “There is no conclusion in the article, just a mention in the Limitations section.”. The manuscript makes several conclusions on the basis of variables that predict treatment-seeking, and the effect on disability outcomes, over the total illness duration.

Reviewer 2 Report

Thank you for sharing this interesting paper investigating the relationship between depression, “life control”, “affective distress”, treatment attendance and perceived disability. Although it is an interesting paper, there are some critical points to be taken into consideration:

·      Patients were not evaluated longitudinally, and the cross-sectional design of the study makes it difficult to name any variables as 'predictors'. Stating in a cross-sectional study which is the cause and which is the effect may be considered an inference

·      There is a lack of information concerning how the “perceived disability” was assessed

·      There is a lack of information on what treatment attendance is evaluated and how it is evaluated: what is the definition of “Attending treatment”? Is it a number of sessions? If yes, which treatment is it?

·      In the Abstract I would write the full name of the scales

·      Paragraph 1 – LBP is described and then CBP is used throughout the manuscript. I assume it means Chronic Back pain, but I can’t find where it is specified

·       "Utilising the Problem Solving Inventory (PSI, third subscale, Heppner & Peter- 69 son, 1982) the found significant correlations problem solving and pain, disability and depression scores in chronic pain sufferers" Maybe there is a typo, but I would clarify this statement

·      Paragraph 1.2  - I would add a definition of Life Control

·      “In a later study Härkäpää et al 78 (1996) that health optimism and control beliefs were significant predictors of improve- 79 ments in functional capacity” maybe a verb is missing in this sentence?

·      Paragraph 1.3 - I would add a definition of Affective Distress

·      Line 89 "Affective distress includes anxiety and depression, and both have similar responses on the physiological system as pain" I would clarify this statement. What do you mean by physiological system?

·      Line 115 “entire course of a CBP diagnosis experience” I would clarify what you mean with this expression

·      Paragraph 2.1 - I would add information about where the participants were recruited

·      Paragraph 2.3 what do you mean by "reply-paid"?

·      I would add among the limitations the lack of information concerning the possible psychiatric diagnosis and pharmacological treatment of the patients

·      I would also add among the limitations a mention on the self-administered nature of the used scales

·      I would add a Conclusion paragraph.

Author Response

Reviewer 2:

  1. Patients were not evaluated longitudinally, and the cross-sectional design of the study makes it difficult to name any variables as 'predictors'. Stating in a cross-sectional study which is the cause and which is the effect may be considered an inference

We appreciate this feedback. The unique contribution of knowledge this study offers to the body of literature on chronic back pain, is the data collected from people retrospectively, about their treatment adherence, predictors of this treatment adherence, covering the total duration of their illness. For most participants, the illness was longer than 2 years, with the next most common category of 6 months-2 years. Participants were asked to respond to the key variable of interest – the duration of pain and treatment attendance – over the entire time period of their illness. While we construe predictors, indirect effects and outcomes in the path model, we do not intend to make causal claims about the conclusions. We reviewed the language to ensure that no causal conclusions are made. We acknowledge the limitation of the cross-sectional design in the limitation section. Ideally, prospective longitudinal studies will add an important contribution to this field. However, at present, to the author’s knowledge, this study is unique in examining the psychological predictors of treatment attendance, and subsequent disability measured at the time of the data collection – which for most participants covers at least 6 months+ of chronic back pain. Chronic back is defined in Australia as either 3 months, or 6 months, or persistent pain, and hence the participants meet this criteria.

  1. There is a lack of information concerning how the “perceived disability” was assessed

As stated in the Materials section, we used the Oswestry Low Back Pain Disability Questionnaire (ODQ) to assess disability, at the time of data collection (which in most cases represents people who have experienced back pain for a substantial period of time, as above). This scale is considered the ‘gold standard’ of assessment in the case of lower back pain, in assess functional capacity in the case of lower back pain suffers. We detail this in measures section of the method, including the internal reliability of the scale with this sample, and do not make any changes in response to this.

  1. There is a lack of information on what treatment attendance is evaluated and how it is evaluated: what is the definition of “Attending treatment”? Is it a number of sessions? If yes, which treatment is it?

Treatment attendance was defined as any treatment session with a health professional for the purpose of treating the chronic back pain condition, and any associated symptoms. We have expanded the explanation of this variable in section 2.2.5: Demographic Data Collection Questionnaire (DDCQ).

  1. In the Abstract I would write the full name of the scales

Thank you. We have now included the full names of the scale, and made it clearer that depression, life control and affective distress were tested as indirect predictors of disability, via treatment attendance.

  1. Paragraph 1 – LBP is described and then CBP is used throughout the manuscript. I assume it means Chronic Back pain, but I can’t find where it is specified

Thank you, we have now included the full wording of chronic back pain (CBP) in the first instance.

  1. "Utilising the Problem Solving Inventory (PSI, third subscale, Heppner & Peter- 69 son, 1982) the found significant correlations problem solving and pain, disability and depression scores in chronic pain sufferers" Maybe there is a typo, but I would clarify this statement.

Thank you. Corrected and rephrased:” they found significant correlations between problem solving and pain, disability and depression scores in chronic pain sufferers.”

  1. Paragraph 1.2  - I would add a definition of Life Control

Thank You. Definition added (Pietila, 1998).

  1. In a later study Härkäpää et al (1996) that health optimism and control beliefs were significant predictors of improvements in functional capacity” maybe a verb is missing in this sentence?

Thank you. Added “found” (86).

  1. Paragraph 1.3 - I would add a definition of Affective Distress

Thank you definition added.

  1. Line 89 "Affective distress includes anxiety and depression, and both have similar responses on the physiological system as pain" I would clarify this statement. What do you mean by physiological system?

Thank you. Added: Shared neurological pathways

  1. Line 115 “entire course of a CBP diagnosis experience” I would clarify what you mean with this expression

Thank you. Added: “the experience of back pain beyond the six months period”.

  1. Paragraph 2.1 - I would add information about where the participants were recruited

Thank you. Added: “with the assistance of health professionals, who distributed the questionnaires amongst their patients” (138)

  1. Paragraph 2.3 what do you mean by "reply-paid"?

Added: return postage provided (182).

  1. I would add among the limitations the lack of information concerning the possible psychiatric diagnosis and pharmacological treatment of the patients

     I would also add among the limitations a mention on the self-administered nature of the used scales

Thank you. Added: “This study did not collect data on participants’ psychiatric diagnoses or pharmacological treatment, further studies may consider these two aspects and include them for further investigation. In addition, the study relied completely in self-reported data which must be considered while evaluation the findings. (415-419).

  1. I would add a Conclusion paragraph.

Thank you. Rephrased and restructured the last few paragraphs.

Reviewer 3 Report

Here are my comments:

1. 1. Introduction: The author(s) need to update its references; the majority are from 200X to 201X. 

2. 1.4. Aims and hypotheses: The author(s) said, "There is limited research about ..." but your ref is 2004? Now it is 2023, more than 19 years. Are you sure this is correct?

3. 2.  Method: the author(s) have to provide power analysis. It would be best to use a path model (because you are using path analysis).

4. I2.2.2. "... coefficient of 0.75.r, internal consistency ..." Please correct the typos.

5. 2.3, the response rate is 44.66%, is this classified as low or moderate in your country on this survey? What is your sampling method? Is it face-to-face or self-reported?

6. 2.5. Can you explain how this hypothesized model (Figure 1) was developed in more detail? For example, PS directly impacts Dis, and will PS affect Dis via Dep (r=0.35**) or AD (r=0.38**)?

7. 3. Results: Good and comprehensive.

8. 4. Discussion: Very comprehensive.

9. 5. Limitations: Please add how the low/moderate response rate will affect the results. Also, any paragraph mentioned the implications of the model in the clinical setting.

Author Response

  1. 1. Introduction: The author(s) need to update its references; the majority are from 200X to 201X.

Thank you. Several references have been updated and replaced with more recent studies.

  1. 1.4. Aims and hypotheses: The author(s) said, "There is limited research about ..." but your ref is 2004? Now it is 2023, more than 19 years. Are you sure this is correct?

Thank you. To the authors best knowledge there are no other studies that explore the impact of [psychological factors on treatment adherence.

  1. 2.  Method: the author(s) have to provide power analysis. It would be best to use a path model (because you are using path analysis).

Thank you. We are using a path model (multivariate). Added information for clarification (results’ section-sample size requirements).

  1. I2.2.2. "... coefficient of 0.75.r, internal consistency ..." Please correct the typos.

Corrected, thank you.

  1. 2.3, the response rate is 44.66%, is this classified as low or moderate in your country on this survey? What is your sampling method? Is it face-to-face or self-reported?

Thank you. Additional information included. Wu et al. (2022) indicated that the average online survey response rate is 44.1%.

  1. 2.5. Can you explain how this hypothesized model (Figure 1) was developed in more detail? For example, PS directly impacts Dis, and will PS affect Dis via Dep (r=0.35**) or AD (r=0.38**)?

Thank you. Pain severity and Pain duration are used as control variables in this model. We are not measuring the effects (direct/indirect) on treatment engagement or disability.

  1. 3. Results: Good and comprehensive.

Thank you.

  1. 4. Discussion: Very comprehensive.

Thank you.

  1. 5. Limitations: Please add how the low/moderate response rate will affect the results. Also, any paragraph mentioned the implications of the model in the clinical setting.

Thank you. Additional information added in relation to clinical implications of the findings.

The authors would like to thanbk the reviewers for they comments which have improve the clarifty and quality of this manuscript. 

Round 2

Reviewer 3 Report

The authors have changed the MS, in particular, and added the key sections of the replies to reviewers into the MS. Well done!